# The *bZIP* Transcription Factors in Current Jasmine Genomes: Identification, Characterization, Evolution and Expressions

**DOI:** 10.3390/ijms25010488

**Published:** 2023-12-29

**Authors:** Kai Zhao, Xianmei Luo, Mingli Shen, Wen Lei, Siqing Lin, Yingxuan Lin, Hongyan Sun, Sagheer Ahmad, Guohong Wang, Zhong-Jian Liu

**Affiliations:** 1College of Life Sciences, Fujian Normal University, Fuzhou 350117, China; zhaokai@fjnu.edu.cn (K.Z.); luoxianmei1008@163.com (X.L.); paramountshen@163.com (M.S.); 15159367350@163.com (W.L.); 19305061020@163.com (S.L.); lyx2003030629@163.com (Y.L.); 13035614038@163.com (H.S.); 2Ornamental Plant Germplasm Resources Innovation & Engineering Application Research Center, Key Laboratory of National Forestry and Grassland Administration for Orchid Conservation and Utilization, College of Landscape Architecture and Art, Fujian Agriculture and Forestry University, Fuzhou 350002, China; sagheerhortii@gmail.com

**Keywords:** jasmine cultivars, stress response, individual genomic differences, evolution, functional differentiation

## Abstract

Jasmine, a recently domesticated shrub, is renowned for its use as a key ingredient in floral tea and its captivating fragrance, showcasing significant ornamental and economic value. When cultivated to subtropical zone, a significant abiotic stress adaptability occurs among different jasmine varieties, leading to huge flower production changes and plantlet survival. The bZIP transcription factors (TFs) are reported to play indispensable roles in abiotic stress tolerance. Here, we performed a genome-level comparison of *bZIP*s using three-type jasmine genomes. Based on their physicochemical properties, conserved motif analysis and phylogenetic analysis, about 63 *bZIP* genes were identified and clustered in jasmine genomes, noting a difference of one member compared to the other two types of jasmines. The *HTbZIP* genes were categorized into 12 subfamilies compared with *A. thaliana*. In cis-acting element analysis, all genes contained light-responsive elements. The abscisic acid response element (ABRE) was the most abundant in *HTbZIP62* promoter, followed by *HTbZIP33*. Tissue-specific genes of the *bZIPs* may play a crucial role in regulating the development of jasmine organs and tissues, with *HTbZIP36* showing the most significant expressions in roots. Combined with complicated protein interactions, *HTbZIP62* and *HTbZIP33* might play a crucial role in the ABA signaling pathway and stress tolerance. Combined with RT-qPCR analysis, *SJbZIP37/57/62* were more sensitive to ABA response genes compared with other *bZIPs* in DJ amd HT genomes. Our findings provide a useful resource for further research on the regulation of key genes to improve abiotic stress tolerance in jasmine.

## 1. Introduction

Jasmine (*Jasminum sambac* (L.) Aition) is prized for its elegant fragrance. Beyond its role as an ornamental flower, it serves a great purpose as tea and spice. Renowned for its significance in tea production and flavor enhancement, jasmine holds significant importance for its ornamental, medicinal and economic attributes [1]. Based on the number of petals, jasmine can be divided into single-petal, double-petal and multi-petal types [2]. Hutou jasmine, also called *Jasminum sambac* var. Grand Duke of Tuscany, produces multi-petal flowers with over 50 petals, which is an important breeding trait used to enhance the ornamental value of new cultivars [3]. Jasmine tea has gained historical significance in Fujian province of China, where the environmental conditions strongly favor the optimal growth of jasmine, especially in the Fuzhou City [4]. With the acceleration of global climate change, Fuzhou has experienced a significant climate change in temperature. Simultaneously, the number of low-temperature days has declined, accompanied by an increase in extreme low-temperature events. Therefore, the effects of abiotic stresses such as low temperature and drought, leading to the reduced flowering and low production of jasmine, have received more attention.

The basic leucine zipper (bZIP) transcription factors (TFs) are widely distributed in plants that play a major role in controlling the various responses against external stimuli [5]. The bZIPs exhibit two distinct structural characteristics: a highly conserved DNA-binding region and a leucine zipper dimer region that exhibits variability in different species [6]. The DNA-binding region typically consists of 16 amino acid residues, characterized by a highly conserved N-X7-R/K motif. The leucine zipper region is composed of seven amino acid residues forming a repeat unit, with leucine or related hydrophobic amino acids at seventh amino acid position. The number of repeat units ranges from three to eight [6,7,8]. Based on the two structures, the bZIP domain can form an α-helix. The α-helix of the two bZIP proteins is staggered to form a zipper structure. Following the formation of homodimers or heterodimers, bZIP proteins can effectively perform their functions [9]. The bZIP TFs can be induced by different stressors, such as drought, salinity and cold. They can interact with the promoter region of related genes to regulate the transcription level of target genes and then regulate the stress tolerance of plants [10]. There are two types of the bZIP TFs involved in low temperatures, ABA-dependent and non-ABA-dependent [11]. The ABA-independent TFs sense low-temperature-induced oxidative stress and respond by interacting with ethylene response factors (ERFs) to modulate downstream cold-responsive genes. In contrast, the ABA-dependent TFs regulate cold-responsive (COR) genes in the ABA signaling cascades. Specifically, upon ABA stimulation, bZIP TFs could bind to ACGT motifs in the promoters of ABA-inducible genes to control the expression of downstream targets within the COR regulon [12,13,14]. For example, *bZIP* TFs in japonica rice [15] and wheat [16] rely on ABA to participate in the regulation of low-temperature stress and play a protective and defensive role.

Today, the increasing number of whole genome sequencing has facilitated the identification of *bZIP* gene family members in a variety of crops; for example, 78 bZIPs are identified in *A. thaliana* [7], 89 in rice [17], 125 in maize [18], 55 in grape [19] 120 in apple [20] and 69 in tomato [21]. However, little is documented on the bZIP members in jasmine. Therefore, we identified *bZIP* genes in three jasmine genomes and analyzed the bZIP TFs by structure characters and relative expression profiles. The findings of this study can be a useful genetic tool in the further molecular breeding of jasmines toward rapid climate change and the domestication mechanism to low-temperature stress.

## 2. Results

### 2.1. Gene Identification and Proteins Characteristics of the bZIP Gene Family

The bZIP_1 (ID: PF00170), bZIP_2 (ID: PF07716) and bZIP_Maf (ID: PF03131) were used to search in the genome databases using BLAST analysis in TBtools v2.030. After eliminating the genes with incomplete structural domains, we identified a total of 64 *JsbZIPs* in single-petal jasmine (abbreviated as *SJbZIPs*), 63 *JsbZIPs* in double-petal jasmine (abbreviated as *DJbZIPs*) and 63 *JsbZIPs* in multi-petal jasmine (abbreviated as *HTbZIPs*) (Figure 1). The genes were named according to their localization on the chromosomes (Appendix A). The amino acid lengths of the three jasmine bZIP proteins showed significant variations. The longest amino acid length was 778 aa (*SJbZIP09*, *DJbZIP10* and *HTbZIP10*), and the shortest amino acids included 112 aa (*SJbZIP29*), 132 aa (D*JbZIP41*) and 110 aa (*HTbZIP51*). The relative molecular weights (WM) ranged from 13.33 kDa (*SJbZIP32*) to 84.10 kDa (*SJbZIP09*), from 15.56 kDa (*DJbZIP05*) to 84.11 kDa (*DJbZIP10*) and from 12.75 kDa (*HTbZIP57*) to 84.11 kDa (*HTbZIP10*) in the three jasmines, respectively. Meanwhile, the isoelectric points (pI) of the *SJbZIP*, *DJbZIP* and *HTbZIP* proteins did not show differences, and the maximum pI was 10.55, with a consistent range from 4.77 to 10.55 for both single- and double-petal jasmine (Figure 1C). The *SJbZIP*s contained 31 basic proteins, the *DJbZIP*s contained 29 basic proteins and the *HTbZIPs* exhibited 30 basic proteins. Additionally, we found that the Grand Average of Hydrocity (GRAVY) of the bZIP proteins of three jasmines was negative, and all of them were hydrophilic proteins (Appendix A).

### 2.2. Phylogenetic Tree Analysis of bZIP Genes of Jasminum sambac

The *SJbZIP*, *DJbZIP* and *HTbZIP* groups were differentiated according to the *AtbZIP* family grouping using a phylogenetic approach. AtbZIP proteins were divided into 13 groups, and the three jasmine *bZIP* gene families were divided into 12 groups, lacking the M group (Figure 1A). The S groups (17 genes of *SJbZIP*, 16 genes of *DJbZIP* and 18 genes of *HTbZIP*) had the highest number of *bZIP* gene family members, followed by the A groups (14 genes). The B, H and J groups contained the lowest number of *bZIP* gene family members, each containing one gene (Figure 1B).

### 2.3. Analysis of Conserved Domain and bZIP Genes Structure

Based on MEME online website and TBtools software, we analyzed motif and exon-intron structures of *HTbZIP* genes (Figure 2). All of the HTbZIP proteins contained motif 1, which was a typical conserved motif (Figure 2E). Members of the F, H, C and G groups contained only motif 1 and motif 4, whereas the K group showed only one conserved motif (motif 1). Notably, the distribution of motifs was specific. For example, motif 6 and motif 5 existed only within the A group, and motif 12 existed only within the S group (but not all members contained it). HTbZIP proteins of the D group contained 6~8 conserved motifs, wherein motif 11, motif 2, motif 8, motif 14 and motif 3 existed only in the D group (Figure 2A).

We further studied the intron-exon distribution of the *HTbZIP* gene family. All of the *HTbZIP* genes contained two or more structures, except for *HTbZIP17*, *41*, *46* and *59*, which contained only one exon. The D and G groups contained a large number of exons, with short lengths. The number of introns had a range of 8 (only *HbZIP24* and *38*)~15, but *HTbZIP13* did not have UTR structure. We also found that the structure of genes in the same groups were similar. For example, all genes of the S group contained an exon, wherein *HTbZIP05* and *HTbZIP56* contained only an exon. The I group had four exons with a long exon length near the 5′ end, two middle exons with short length and a short exon near the 3′ end (Figure 2B,D). Meanwhile, the comparison of the number of introns in the three jasmine *bZIP* genes revealed that the average number of introns was four, and the minimum number was zero. The *SJbZIP* genes contained a maximum of 12 introns (*SJbZIP44* and *30*, G group), the *DJbZIP* genes contained a maximum of 14 introns (*DJbZIP14*, D group) and the *HTbZIP* genes contained a maximum of 15 introns (*HTbZIP43*, G Group) (Figure 2C).

### 2.4. Target of Specific miRNAs for bZIP Genes

In this study, we selected several members of the *bZIP* gene family in three jasmine species as candidate target gene sequences. The results showed that 26 genes out of the 64 *SJbZIP* genes contained miRNA target sites, totaling 42 miRNA target sites. Among the 63 *DJbZIP* genes, 26 genes contained miRNA target sites, totaling 42 miRNA target sites. Among the 63 *HTbZIP* genes, 26 genes contained miRNA target sites, totaling 43 miRNA target sites (Figure 3B, Appendix A). *SJbZIP18*, *DJbZIP19* and *HTbZIP18* (A group) each contained four miRNA targets, making them the genes with the most miRNA targets among the three jasmine *bZIP* gene families. This was followed by *SJbZIP38/54* (three targets), *DJbZIP37* (three targets) and *HTbZIP37/54* (three targets), all of which belonged to the I group (Figure 1A and Figure 3C). These findings suggested a complex regulatory relationship between miRNAs and the *bZIP* genes of jasmine. Predicting the miRNA targets of various *bZIP* genes was beneficial for further investigating the mutual regulatory relationship between *bZIP* genes of jasmine and miRNAs.

### 2.5. Synteny Analysis of HTbZIP Genes

The position information of the *HTbZIP* genes on the chromosome was extracted according to the gff3 file, and the localization of the *HTbZIPs* on the genome chromosome was visualized by TBtools. The results showed that hutou jasmine had a total of 13 chromosomes, and 63 *HTbZIP* genes were unevenly distributed on 13 chromosomes (Figure 4A), with the least containing 1 *HTbZIP* gene (Chr 11) and the most containing 8 *HTbZIP* genes (Chr 07 and Chr 10). The distribution of *HTbZIP* genes on chromosomes also corresponds to gene density. Among them, *HTbZIP03*, *10*, *21* and *29* were distributed in a low-density region, and the rest were distributed in a higher-density region.

Based on MCScanX analysis in TBtools software, we analyzed collinearity of *HTbZIP* genes. The results showed a total of 36 pairs of segmental duplication within the species, and two pairs of tandem repeats. Segmental duplication events mainly occurred in Chr 06 and Chr 07, but there was no fragment duplication in Chr 01, Chr 08 and Chr 11. Multiple fragment replication events also occurred in the same gene, such as *HbZIP30*-*HTbZIP46* and *HbZIP30*-*HTbZIP57*. Tandem repeats occurred only in Chr 04 and Chr 12, representing *HTbZIP15* and *HTbZIP16* (A group), *HTbZIP58* and *HTbZIP59* (S group), respectively (Figure 4A). Meanwhile, the collinear analysis between species showed that there were 63 homologous gene pairs between *HTbZIPs* and *A. thaliana*, 105 homology pairs between *HTbZIPs* and single-petal jasmine and 104 homology gene pairs between *HTbZIPs* and double-petal jasmine. Neither hutou jasmine nor *A. thaliana* had a homologous *bZIP* gene between Chr08 and Chr11, but both hutou jasmine and single- and double-petal jasmine had homologous genes (Figure 4B). It was worth noting that the collinearity between hutou jasmine and two jasmines was higher than that between *A. thaliana* varieties, but the collinearity between the three jasmine species was similar, reflecting the evolutionary differences between different species and the correlation between different subspecies. These data revealed that the *bZIP* genes exhibited the same evolutionary pattern across various species and subfamilies.

### 2.6. Analysis of Codon Usage Bias of HTbZIP Genes

The results of codon preference analysis using condonW and EMBOSS showed that the average U3 content was 39.09%, the average C3 content was 22.65% and the average A3 content was 32.96%. The average content of G3s was 31.31%, and the average content of U3s and A3 was more than that of C3s and G3s, indicating that the *HTbZIP* gene preferentially used synonymous codons ending in U/A. The ENC value of all *HTbZIPs* was higher than 35, the range was 40.42~61. The codon of *HTbZIPs* contained weak preference when encoding proteins, among which *HTbZIP31/34/39* had an ENC value of 61, indicating that each codon of the three genes was used equally. The CAI value rane was 0.131~0.234, and the average value was 0.179, among which the codon of *HTbZIP33* gene was the most adaptable. The Fop value range was 0.323~0.484, and the average value was 0.383 (Appendix A, Figure 5A).

In the ENC-plot analysis (Figure 5B), the distribution of *HTbZIPs* was relatively concentrated. A small number of *HTbZIPs* were found on the standard baseline, whereas most of them were under the standard baseline. Their codon preference was mainly affected by natural selection. The analysis of pR2 showed that *HTbZIPs* were evenly distributed around the intersection point (0.5, 0.5), and the degree and direction of bias indicated that codon preference was affected by mutational pressure and natural selection (Figure 5C). In the neutrality-plot analysis, the distribution range of GC12 was 0.362–0.554, and the distribution range of GC3 was 0.333–0.578. The correlation between GC3 and GC12 was r = 0.18, and the regression coefficient was −0.144, which showed no significant negative correlation, indicating that the preference for codon usage was mainly affected by natural selection (Figure 5D). At the same time, a total of 26 *HTbZIP* RSCUs were higher than 1, exhibiting a high frequency of use. Among them, AGA (1.67) had the highest RSCU value and the highest codon preference, and CGC (0.39) had the lowest frequency of use, with almost no codon preference (Figure 5E).

### 2.7. Prediction of Cis-Acting Elements of Promoter in HTbZIP Genes

TBtools software was used to extract 1000 bp sequences upstream of the coding region of the *HTbZIP* genes, and then the sequences were submitted to the online website PlantCARE for the prediction and statistics of cis-acting elements. The results showed that light-responsiveness elements existed in all *HTbZIP* genes. These were among the most abundant response elements, accounting for 46.18% (363/786) of the total number of elements counted. G-Box elements were the most widely distributed, accounting for 26.17% (95/363) of the total number of light-responsiveness elements. Among the 11 phytohormone responsive elements, the ABRE element of abscisic acid (ABA) was the most abundant. *HTbZIP62* (7) showed the most ABRE elements. ABRE were followed by TGACG-motif and CGTCA-motif element of MeJA-responsiveness. Among the six environmental stress responsive elements, ARE (AAACCA) elements of anaerobic induction were the most abundant, followed by the MBS (CAACTG) element of drought response, and the LTR (CAACTG) elements of low-temperature response. Plant growth responsive elements were the least distributed, accounting for only 6.36% (50/786) of the total number of components. At the same time, *HTbZIP62* (29 elements) exhibited the largest number of elements, followed by *HTbZIP33* (24 elements). *HTbZIP24* contained only three elements, which was the lowest (Figure 6A). The 0~35 bp sequence range contained all elements except for the MeJA hormone response element, CGTCA-motif and TGACG-motif. The other four elements were present but in small numbers. The distribution of numbers for the elements mainly fell within the range of 35~500 bp and 500~1000 bp. The number distribution of G-Box, ABRE and ARE elements within a range of 500~1000 bp was exceeded within the range of 35~500 bp (Figure 6B).

### 2.8. Expression Profiles of HTbZIP Genes in Different Tissues

Based on the transcriptome data, we analyzed the differential expression patterns of the *bZIP* genes in jasmine across the roots, stems, leaves and flowers (Figure 7). In hutou jasmine, *HTbZIP27* (belonging to the K group) was highly expressed in all tissues, with particularly high expression in the roots. The expression of *HTbZIP36* (S group) was significantly higher in both roots and leaves. We also found that *HTbZIP05* (group S), *10* (group B), *32* (group I) and *22* (group I) were highly expressed in the roots, stems, and leaves. *HTbZIP56* (S group) was highly expressed in three periods of flower development (Figure 7A). Combined with gene expression trend analysis, *HTbZIP36* exhibited the highest expression in roots (Cluster 9), while *HTbZIP27* showed the high expression in expression in roots (Cluster 8). *HTbZIP06* (D group)/31 (S group)*/52* (S group)*/26* (S group) (cluster 5) showed a high expression trend only in three periods of flower development. *HTbZIP05/10/32/22* (cluster 7) exhibited the highest expression in roots (Figure 7B). In single-petaled jasmine, the *SJbZIP53*/57 (S group) were highly expressed in both roots and stems. *SJbZIP19* (S group) was highly expressed in the stem and during the third flowering stage. *SJbZIP24* (D group) was highly expressed only exclusively in roots (Figure 7D). At the same time, *DJbZIP36* (S group) was highly expressed in the roots of double-petal jasmine. *DJbZIP28* (K group)/*53* (I group)*/23* (I group)*/58* (S group)*/05* (S group) were relatively highly expressed. *DJbZIP52/20* (S group) were highly expressed in the stems, *and DJbZIP20/36* were also expressed during the third flowering stage (Figure 7E). There was no gene with high expression of the *bZIP* genes in the leaves of single- and double-petal jasmine. However, there were genes with high expression of the *bZIP* genes in the leaves of hutou jasmine.

In the five flowering stages of single-petal jasmine, *SJbZIP53* was expressed only at the F5 stage. *SJbZIP19/57/04* (S group) exhibited consistently high or elevated expression levels throughout the flowering period, while *SJbZIP57/04* showed high expression levels at the F5 stage (Figure 8A). During the entire flowering period of double-petal jasmine, *DJbZIP20/36* (group S) consistently exhibited high levels of expression. However, there was variation in the specific stages of flowering, with high levels of expression. D*JbZIP20* showed high expression during the F1~F4 period and showed high expression levels expressed in the F2 and F3 stages. *DJbZIP58/52* (group S) were highly expressed only during the F5 period. Furthermore, in addition to the regulation of the S group in jasmine during the flowering stage, there was also a notable high expression of an E group member (*DJbZIP14*) at the F5 flowering stage (Figure 8B). During the various flowering stages of hutou jasmine, the expression levels of *HTbZIP06* in F3~F5 were higher, while the expression levels of other genes were lower in the D group. The expression levels of *HTbZIP27* were higher in all five flowering stages. At the same time, 33.33% (6/18) of the members of the S group of *HTbZIP* genes were highly or relatively expressed, and they were involved in regulating the flowering period. Among them, *HTbZIP56* maintained high expression throughout the flowering period, especially during the F3 and F4 stages. The expressions of *HTbZIP52/36* were relatively high throughout the entire flowering period. In the F3~F5 stage, *HTbZIP31* was highly expressed. However, the expression levels of the remaining members of the C/G/H/A/F group were moderate or low. These results suggeste that tissue-specific genes of the *bZIP* genes family may play a crucial role in regulating the development of jasmine organs and tissues.

### 2.9. Analysis of the Protein–Protein Interaction Network of HTbZIPs

We used STRING 12.0 to construct a protein–protein interaction network of 63 HTbZIP proteins. The results showed that DPBF3 (*HTbZIP07*/*15*/*16*/*47*/*60*), ABI5 (*HTbZIP11*/*63*), TGA2 (*HTbZIP02*/*13*), TGA6 (*HTbZIP06*), TGA3 (*HTbZIP38*), TGA4 (*HTbZIP24*), ABF2 (*HTbZIP28*/*5*) and ABF4 (*HTbZIP18*/*51*) were involved in abscisic acid-regulated gene expression. BZIP44 (*HTbZIP19*/*36*) was involved in seed germination. TGA9 (*HTbZIP04*/*48*) and TGA10 (*HTbZIP61*) were bZIP transcription factors required for another development. PAN (*HTbZIP45*) was involved in flower organ formation, HY5 (*HTbZIP42*) was involved in cryptochrome signaling transduction and FD (*HTbZIP17*) promoted excessive flowering. Additionally, BZIP19 (*HTbZIP20*/*54*) was involved in the response to zinc iron deficiency (Figure 9). These results indicated that the HTbZIP proteins had diverse functions and played an important role in the growth and development of jasmine.

### 2.10. Response Profiles of HTbZIP Genes with ABA Treatments

In this study, the roots, leaves and flowers of hutou jasmine and the *HTbZIP* genes containing more ABRE elements and their corresponding *SJ/DJbZIP* genes were selected as the target genes, as these genes may respond to the regulation of hormone expression under ABA treatment. RT-qPCR was used to analyze the response expression patterns of these *bZIP* genes after ABA treatment. The results showed that the highest expression level of *SJbZIP05*/*13* in single-petal jasmine was treated with ABA for 12 h and then decreased. The expression levels of *SJbZIP37*/*62* showed an upward trend in ABA treatment for 0~24 h. The expression pattern of *SJbZIP57* was different from other genes. After 6 h of ABA treatment, the expression level decreased to a lower level than the control group and then increased with the increase of treatment time (Figure 10A). In the double-petal jasmine, the expression levels of *DJbZIP06*/*14*/*28* were the highest after ABA treatment for 12 h, and then the expression level showed a downward trend. The expression levels of *DJbZIP36*/*63* were lower than that of the control group after 6 h of ABA treatment, and the expression trend first decreased, and then increased and decreased (Figure 10B). In hutou jasmine, the expression patterns of *HTbZIP06*/*56*/*36* were similar under ABA treatment. *HTbZIP27* expression increased with ABA treatment time (Figure 10C). It showed a trend of the first increasing and then decreasing, but the expression level was higher than other treatment groups at ABA treatment for 12 h (Figure 10C). The response of these genes to ABA varied, and the expression levels of *SJbZIP37/57/62* were much higher than those of other genes after 12 and 24 h of ABA treatment. It is speculated that three genes may be involved in the response to ABA.

## 3. Discussion

The *bZIP* genes are closely related to plant growth and development and stress tolerance. We analyzed the *bZIP* gene families of single- and double-petal jasmine and hutou jasmine at the genome-wide level. We identified 64 *bZIP* genes in single-petal jasmine, 63 *bZIP* genes in double-petal jasmine and 63 *bZIP* genes in hutou jasmine. This gene count is comparable to the other species, such as *Chrysanthemum indicum* (66 genes) [22], *Nelumbo nucifera* (59 genes), *Nelumbo lutea* (65 genes) [23] and tomato (69 genes) [21]. However, the gene count is lower than that of *A. thaliana* (78 genes) [7], potato (104 genes) [24] and maize (125 genes) [18]. Notably, the number of *bZIP* genes remained consistent across jasmine subspecies, and there were no significant differences in the physicochemical properties of the proteins. Unlike *A. thaliana*, which was divided into 13 groups, the three kinds of jasmines were divided into 12 groups. They lacked an M group, which was supposed to be lost during the evolutionary process.

In the conserved motif and gene structure analyses, the conserved motifs of *HTbZIP* genes in the same group were similar. All of them had one conserved motif (motif01), while the different groups possessed specific conserved motifs, suggesting that the *bZIP* genes were conserved during the course of evolution. Meanwhile, the distribution of introns and exons in the gene structure of different groups displayed specific features. For example, the S group contained only one exon, which was similar to studies of other species, such as cassava [25].

Covariance analysis showed that there were 2 tandem repeat events and 36 segmental duplication events in hutou jasmine. Therefore, segmental duplication might be the main pathway for the expansion of the *HTbZIP* genes. The interspecies covariance revealed 63 homologous gene pairs between the hutou jasmine and *A. thaliana*, 104 homologous gene pairs with single-petal jasmine and 103 homologous gene pairs with double-petal jasmine. The number of homologous gene pairs between subspecies was much higher than that of *A. thaliana*, but the comparison of homologous gene pairs between subspecies differed by only one, suggesting the consistency of subspecies evolution. Meanwhile, the *HTbZIP* genes first used synonymous codons ending in A/U, which was the same as the *JrbZIP* genes [26]. Codon preference affected the elongation rate and folding process of proteins, but codon preference was also affected by base composition, natural selection and mutational pressure [27,28]. In this study, both ENC- and neutrality-plot analysis showed that the codon usage bias of *HTbZIP* genes was mainly affected by natural selection and partly or solely by mutational pressure.

The 1000 bp sequence upstream of the *HTbZIP* gene was extracted and analyzed for cis-acting elements. All genes contained light-responsive elements, which is compatible with jasmine’s light-loving habit [29]. In RT-qPCR analysis, the expression levels of *SJbZIP37/57/62* all exhibited significant changes in ABA response. They each contained 1–2 ABRE elements (Appendix A), which was speculated to be one of the response genes of ABA.

Many studies have demonstrated that the *bZIP* gene family of transcription factors plays a crucial role in the growth and development of plants [30,31]. In this study, the varying expression levels of the *bZIP* genes in four tissues of three different jasmine varieties suggested that the *bZIP* genes were involved in the growth and development of jasmine (Figure 7). In *A. thaliana*, *AtbZIP11* (S group) negatively regulates root development, and it involves a variety of metabolic pathways [32,33,34]. *HTbZIP36*, *SJbZIP53* and *DJbZIP36* were all members of group S and exhibited high expression levels in the root. It was speculated that they had similar functions to *AtbZIP11*. Similarly, members of the K group (e.g., *AtbZIP60*) mainly act in the endoplasmic reticulum stress pathway [35]. The predictive function of *HTbZIP27* corresponded to that of *AtbZIP60* (Figure 9), and its expression was significant in all tissues. Walnut *bZIP* family S group genes, *JrbZIP20*, *JrbZIP62* and *JrbZIP69*, were highly expressed in early female flower bud differentiation [26]. During the blooming of jasmine flowers, the genes with high expression were mainly distributed in the S group. For example, *SJbZIP19* was highly expressed at the F1~F4 flowering stages, and *SJbZIP53*/*57*/*04* were highly expressed at the F5 flowering stage. *DJbZIP58*/*52* was highly expressed in the F5 stage, while *DJbZIP36*/*20* was highly expressed in the F2 and F3 stages. The expression levels of *HTbZIP31* were high in the F3~F5 phases, and *HTbZIP56*/*52* were highly expressed in F1~F5 phases. The highly expressed genes in specific tissues of hutou jasmine were selected as the target genes for the ABA response assay. It was found that the expression levels of the *SJbZIP57, DJbZIP58* and *HTbZIP56* genes were all influenced by ABA, but the expression trend varied across different tissues.

In conclusion, on the basis of evolutionary relationships, it is speculated that the *bZIP* genes, sharing direct homology or close relation to those in the *A. thaliana* group, may also perform similar functions in jasmine. In addition, some *bZIP* gene groups consist of single-copy or low-copy genes, such as the B, H and K groups. The identification of these groups suggest their functional conservation during plant evolution [36]. As jasmine has evolved from tropical or subtropical origins (such as India and the Persian Gulf) to thrive at relatively higher latitudes (such as Fujian and Guangxi provinces, China), a majority of these varieties have undergone domestication to adapt to low-temperature environments. The *bZIP* gene family could be a potential contributor in promoting the survival of jasmine in its new environment.

## 4. Materials and Methods

### 4.1. Genomics Data and Plant Sources

In this study, the complete genome and the annotated file of single- and double-petal jasmines were utilized from Nation Genomics Data Center (NGDC: PRJCA006739), and the genomics data of hutou jasmine were provided by the inner laboratory of Fujian Normal University. The protein sequences of the *bZIP* gene family members of *A. thaliana* were obtained from Phytozome v13 (https://phytozome-next.jgi.doe.gov/, accessed on 27 September 2023). However, the search for the sequence of *AtbZIP73* did not yield accurate information in the *A. thaliana* genome database and was therefore was not used [7].

### 4.2. The Identification of bZIP Gene Family

The *bZIP* gene family of jasmine was identified by BLASTP and HMMER Search methods in the TBtools v2.030 software [37]. Based on the AtbZIP protein sequences, Markov model with bZIP_1 (ID: PF00170), bZIP_2 (ID: PF07716) and bZIP_Maf (ID: PF03131) from Pfam website [38], we deleted the incomplete domain sequences and finally obtained the jasmine *bZIP* genes, for which the domain analysis used the NCBI-CDD (https://www.ncbi.nlm.nih.gov/cdd/, accessed on 29 September 2023) and Pfam search tools (Pfam: Search Pfam (xfam.org), accessed on 29 September 2023) [39]. Finally, physicochemical properties analyses of bZIP proteins were performed using TBtools v2.030.

### 4.3. Phylogenetic Tree Construction

A phylogenetic tree was constructed with the MEGA 11 software by clustalW alignment and the Neighbour-Joining(NJ) method with predicted protein sequences of jasmine and *A. thaliana* bZIP proteins, with the parameters:bootstrap (1000 replicates), p-distance model and pairwise deletion [40,41]. The classification of jasmine *bZIP* genes were classified according to their phylogenetic relations with their corresponding bZIP genes of *A. thaliana*, the phylogenetic tree was modified using the online website iTOL (https://itol.embl.de/, accessed on 30 September 2023).

### 4.4. Conservative Motifs, Gene Structure of HTbZIPs

The online website MEME (https://meme-suite.org/meme/tools/meme, accessed on 9 October 2023) was used for predicting the conserved motifs of HTbZIP proteins [42], with the number of conserved motifs set to 15 and the other parameters set to default. The ID and GFF files of the *HTbZIP* genes were submitted to the Gene Structure View program of TBtools v2.030 for analyzing the composition of exons and introns. Introns were counted using the GXF-Fix and GXF-Star of TBtools v2.030 and plotted using the GraphPad Prism 9.0.0 (121).

### 4.5. MiRNA Editing for the bZIP Gene Family

Based on the psRNATarget website (https://www.zhaolab.org/psRNATarget/, accessed on 14 December 2023), the miRNA targeting of *bZIP* genes in three jasmine species was predicted. The selected species was *Arabidopsis thaliana*, with the expected value parameter set to 4, while the rest of the parameters were left at their default settings. The targeted genes were aligned with the miRNAs of Arabidopsis, and the results were statistically analyzed. The findings were visualized using TBtools and the GraphPad Prism 9.0.0 (121).

### 4.6. Synteny Analysis of the bZIP Gene Family

The One-Step MCscanX function of TBtools software was utilized to conduct an analysis of *HTbZIP* genes, both intraspecific and interspecific collinearity. The default parameters were used for intraspecific collinearity analysis, while for interspecific collinearity analysis, the E-value was set to 1 × 10^−5^, and the remaining parameters were set to their default values. Intraspecific collinearity analyses encompass various aspects, such as tandem repeat, segmental repeat, gene density and chromosome mapping, which were visually represented in Circos plots. Interspecific collinearity involved examining homologous gene pairs between hutou jasmine and single-petal jasmine, double-petal jasmine and *A. thaliana*, respectively. The above visualizations were created using TBtools v2.030.

### 4.7. Analysis of Codon Usage Pattern of HTbZIP Genes

CondonW1.4.2 was used to analyze the codon usage characteristics of *HTbZIP* genes. The correlation coefficients analyzed were the frequency of optimal codons (FOP), codon adaptation index (CAI), effective number of codons (ENC), GC content of codons and GC content of codons coding for the same amino acid at position 3 (GC3s) and relative synonymous codon usage (RSCU) [43]. GC, GC2 and GC3 were analyzed using the CUPS function of the online website EMBOSS (https://embossgui.sourceforge.net/demo/, accessed on 25 October 2023) [44]. The ENC-plot analysis and stacked barplot of RSCU values were drawn using the R Programming Language. The ENC-plot is a standard plot generated using the formula ENC = 2 + GC3 + 29/[GC3^2^ + (1 − GC3)^2^]. PR2-plot analysis, neutrality-plot analysis, and violin plots were created using GraphPad Prism 9.0.0(121). PR2 was plotted with G3/(G3 + C3) as the *x*-axis and A3/(A3 + T3) as the *y*-axis. Two lines were drawn at X = 0.5 and Y = 0.5 [45]. Neutrality-plot analysis was plotted with the GC3 of CDS for each gene as the abscissa and the mean of GC1 and GC2 (denoted as GC12) as the ordinate.

### 4.8. Cis-Acting Element Analysis of HTbZIP Genes

The 1000 bp upstream sequence of *bZIP* genes in hutou jasmine was obtained by TBtools. Putative cis-regulatory elements within the promoters of the *bZIP* genes were identified using the PlantCARE tool (http://bioinformatics.psb.ugent.be/webtools/plantcare/html/, accessed on 15 October 2023) [46]. After the data were statistically classified, TBtools v2.030 was used for data visualization. The main elements of the three jasmines were drawn using the GraphPad Prism 9.0.0 (121).

### 4.9. Expression Analysis

The transcription data of hutou jasmine were obtained from our own research group. The TPM value for the *HTbZIP*s, the *SJbZIPs* and *DJbZIP* genes in different tissues (root, stem, leaf, flower) of hutou jasmine were obtained [47], and the logarithm of the TPM value was taken to make a heat map by TBtools v2.030. The Mfuzz function of OmicStudio3.6 (https://www.omicstudio.cn/tool, accessed on 10 October 2023) was used to visualize the expression trend of the *HTbZIP* genes. The correlation heatmap of Chiplot (https://www.chiplot.online/correlation_heatmap.html, accessed on 20 December 2023) was used. The parameter of the Mfuzz plot and correlation heatmap settings were all default.

### 4.10. Protein–Protein Interaction Network Analysis of HTbZIPs

The *HTbZIP* protein sequence was uploaded to STRING 12.0 (https://cn.string-db.org/, accessed on 6 December 2023), and *A. thaliana* was selected as the target species. The candidate protein with the highest bitscore value was selected as the final matched *A. thaliana* protein for the next step of protein network construction. Finally, we obtained an optimal protein–protein interaction network.

### 4.11. RT-qPCR Assay during the ABA Treatment

Three jasmine plant samples were collected from Fujian Agriculture and Forestry University, Fuzhou, China, taking 30~60 cm of branches with some old branches and place them in a natural room (20~25 °C) for one week. The treatment methods were exogenous auxin, 0.5 mg/L ABA solution. The leaf surface and leaf back were sprayed with an atomizer until they were fully moist but no droplets remained. At 0 h, 6 h, 12 h and 24 h, the clean leaves treated with ABA were quickly frozen with liquid nitrogen and stored at −80 °C. The total RNA extraction method of plants was based on the method of the E.Z.N.A ^®^Plant RNA Kit (OMEGA, catalog number: R6827). The Hieff^®^ Ⅲ 1st Strand cDNA Synthesis SuperMix for qPCR was used to synthesize cDNA single strands by the reverse transcription of extracted RNA. The cDNA was used as a template, and the Hieff UNICON^®^ Universal Blue qPCR SYBR Green Master Mix kit was used to perform real-time fluorescence quantitative PCR detection (RT-qPCR). The *JsActin* gene of *Jasminum sambac* was selected as the internal reference gene [48]. The reference genes and gene sequence primers used in this experiment are shown in Appendix A. The RT-qPCR reaction was designed as 3 technical replicates with a total system of 20 μL, including 10 μL of Hieff UNICON^®^ Universal Blue qPCR SYBR Green Master Mix, 0.4 μL of Forward primer (10 μM), 0.4 μL of Reverse primer (10 μM), 2 μL of template DNA and 7.2 μL of sterile ultrapure water. The amplification procedure was pre-denaturation at 95 °C for 60 s; 95 °C for 10 s, 60 °C for 30 s, a total of 40 cycles; 95 °C for 5 s and 65 °C for 5 s. The obtained data were calculated using the 2^−ΔΔCt^ calculation method to calculate the relative gene expression levels. ORIGIN (2021) was used for drawing.

## 5. Conclusions

By systematic analysis, *bZIP* genes were identified in three jasmine genomes; there were no large-scale difference in family members, though they were lopsidedly distributed on 13 chromosomes. *bZIP* genes were categorized into 12 groups, while the M group was missing compared with arabidopsis. The structures of different *bZIP* genes were significantly different. Tandem repeats and segmental duplications were observed to varying extents in *HTbZIP*, with segmental duplication events being the major mode of expansion of the *HTbZIP* gene family. However, the difference in promoter led to the expression change, with the *HTbZIP* genes containing several abiotic stress-responsive cis-acting elements, and it is hypothesized that *HTbZIP62* and *HTbZIP14* may be ABA response genes. The differing expression levels of the *bZIP* genes in various tissues of three jasmine species indicate their involvement in the growth and development of jasmine. Combined with real-time fluorescence PCR analysis, the *bZIP* genes may regulate the growth and development of jasmine in response to ABA.

## Figures and Tables

**Figure 1 ijms-25-00488-f001:**
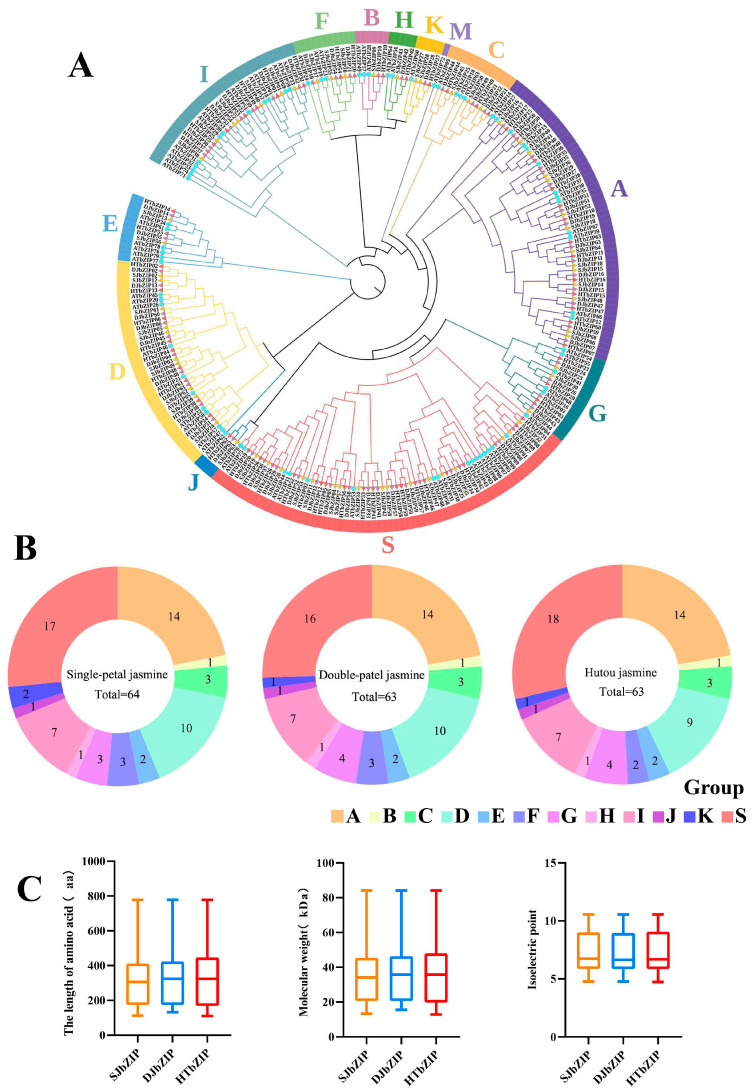
Classifcation of bZIPs in jasmines. (**A**) Phylogenetic tree of bZIP proteins from *A. thaliana*, single- and double-petal jasmine and hutou jasmine. The blue solid circle represents *A. thaliana*, and the different colored solid triangles represent three types of jasmines. (**B**) Pie chart of the bZIPs subfamilies distribution of three jasmines. Different colors represent different subfamilies. (**C**) Physicochemical properties of three types of jasmine.

**Figure 2 ijms-25-00488-f002:**
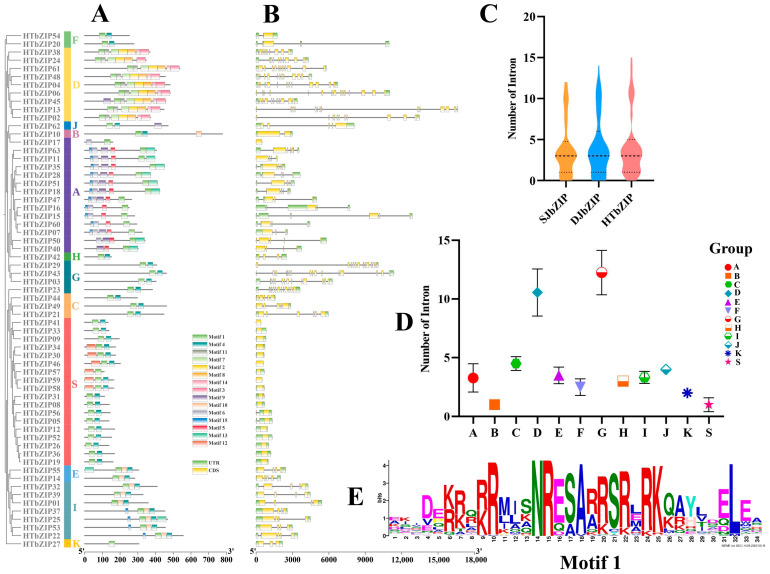
Conserved motif and gene structure of *HTbZIP* genes. (**A**) Conserved motif of *HTbZIP* genes. (**B**) Gene structure of *HTbZIP* genes. (**C**) Comparison of the number of introns in three jasmine *bZIP* genes. (**D**) Comparison of the number of introns between *bZIP* genes groups of hutou jasmine. (**E**) Conserved motif 1 of *HTbZIP* genes.

**Figure 3 ijms-25-00488-f003:**
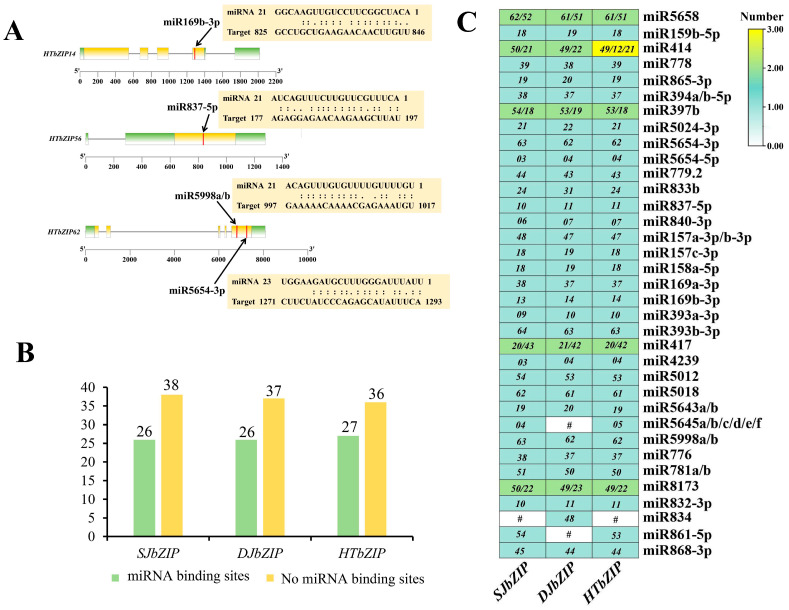
Predicted miRNA target distribution for *bZIP* gens in jasmine. (**A**) Predicted miRNA targets for *HTbZIP14/56/62*. The yellow region represents the coding region (CDS) of the *HTbZIP* genes, and the green region represents the untranslated regions (UTR). (**B**) Statistics of *bZIP* genes containing miRNA targets in three jasmines. (**C**) Statistics of miRNA types predicted by the bZIP gene for three jasmines. The italicized numbers in the grid represent the IDs of the genes. The cell colors indicated the number of miRNA targets.

**Figure 4 ijms-25-00488-f004:**
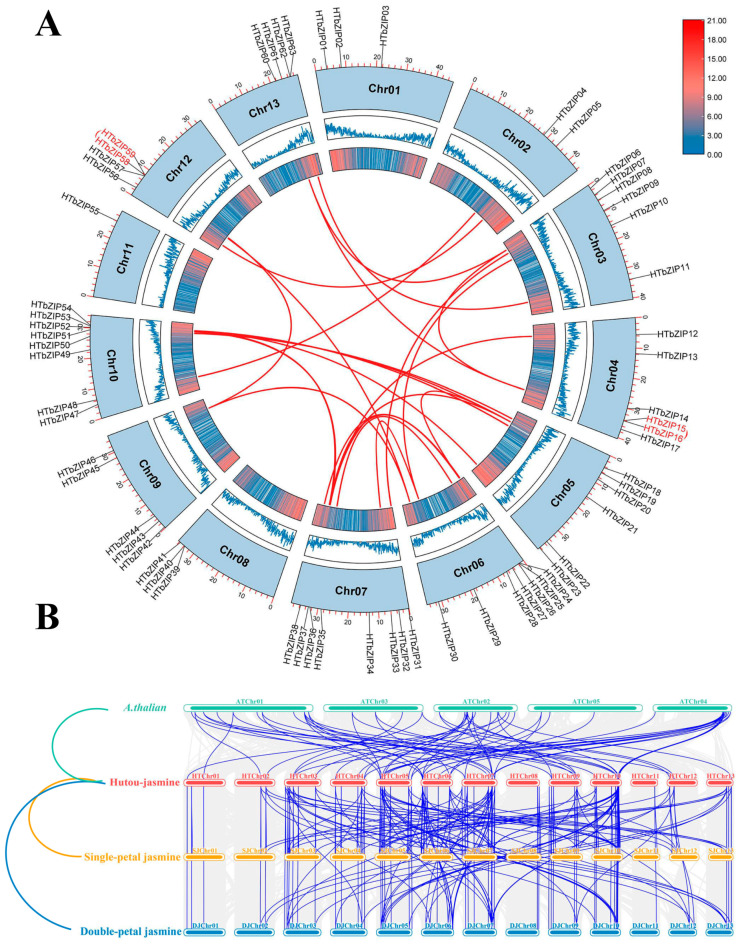
Synteny analysis of the *bZIP* genes in hutou jasmine. (**A**) Intraspecific collinearity. Red font represent tandem repeats. The hat symbol above them means a pair of tandem repeat pairs. Red lines represent segmental duplication. (**B**) Interspecific collinearity. The figure shows the collinearity between hutou jasmine and *A. thaliana*, single-petal jasmine and double-petal jasmine. The short rods of unequal length represent chromosomes, and the blue lines represent homologous pairs between species.

**Figure 5 ijms-25-00488-f005:**
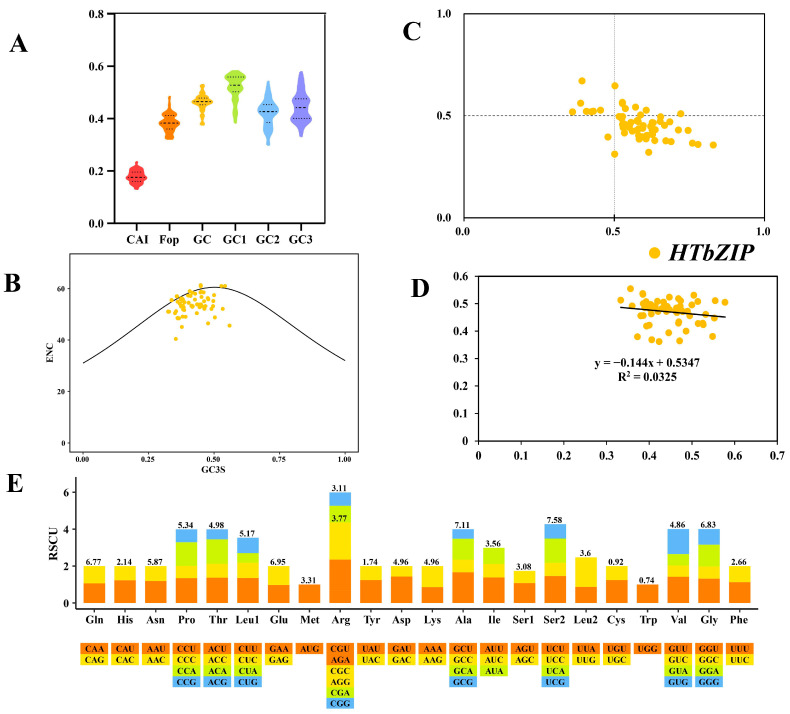
Analysis of codon usage bias of *HTbZIP* genes. (**A**) Codon usage bias parameters of the *HTbZIP* gene. (**B**) ENC-plot analysis of *HTbZIP* genes. (**C**) PR2-plot analysis of *HTbZIP* genes. (**D**) Neutrality-plot analysis of *HTbZIP* genes. GC12 represents the average of GC1 and GC2. (**E**) Frequency of synonymous codon usage of *HTbZIP* genes. The value above the stacked plot represents the frequency of occurrence of this amino acid codon.

**Figure 6 ijms-25-00488-f006:**
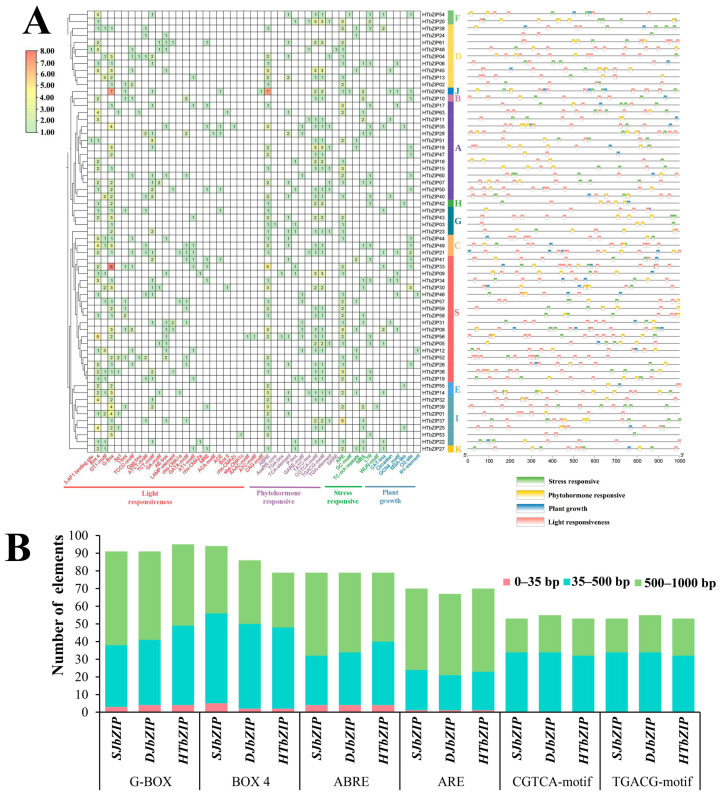
The result of cis-acting elements in promoter of the *HTbZIP* genes. (**A**) Red, yellow/purple, green and blue bars/characters represent the light-responsive, stress-responsive, phytohormone-responsive and plant growth elements in the *HTbZIP* promoter regions, respectively. (**B**) The distribution of the main response elements of the three jasmine *bZIP* genes across different regions.

**Figure 7 ijms-25-00488-f007:**
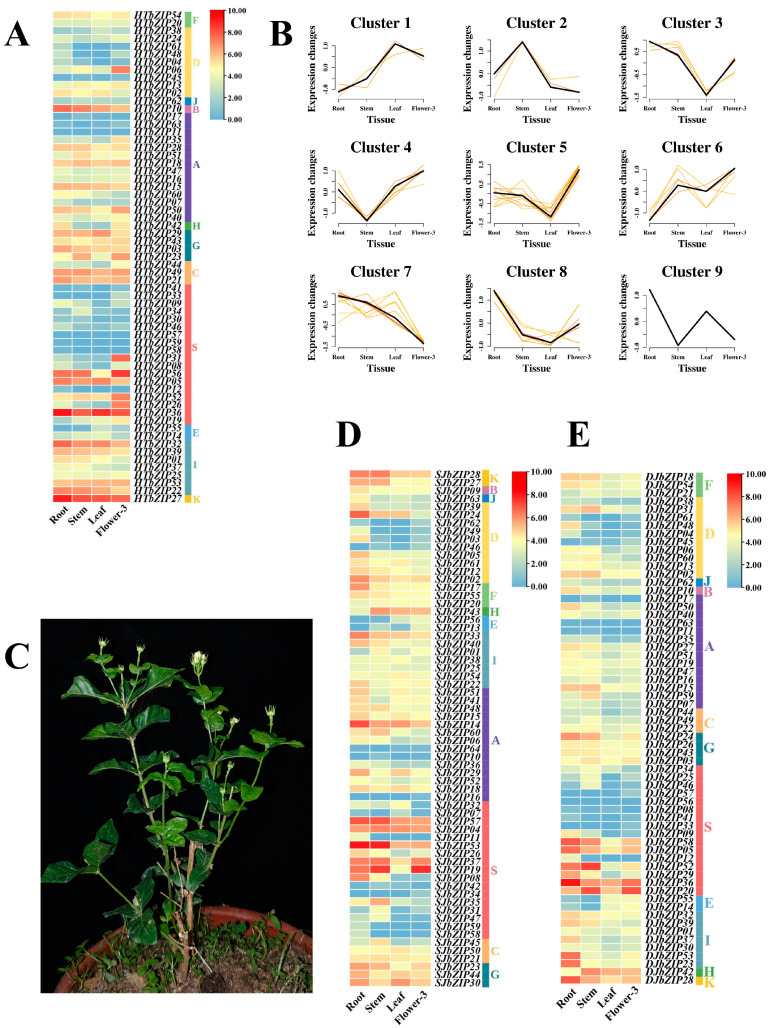
Expression levels of *bZIP* genes with different jasmine tissues. (**A**) Expression levels of *HTbZIP* genes in different tissues of hutou jasmine. (**B**) Expression trend of *HTbZIP* genes in 4 tissues of hutou jasmine. The black lines represent the central trend change and the other color lines represent the trend change results of different genes. (**C**) Hutou jasmine display diagram. The figure includes the flower, stem and leaf tissues. (**D**) Expression trend of *SJbZIP* genes in 4 tissues of hutou jasmine. (**E**) Expression trend of *DJbZIP* genes in 4 tissues of hutou jasmine.

**Figure 8 ijms-25-00488-f008:**
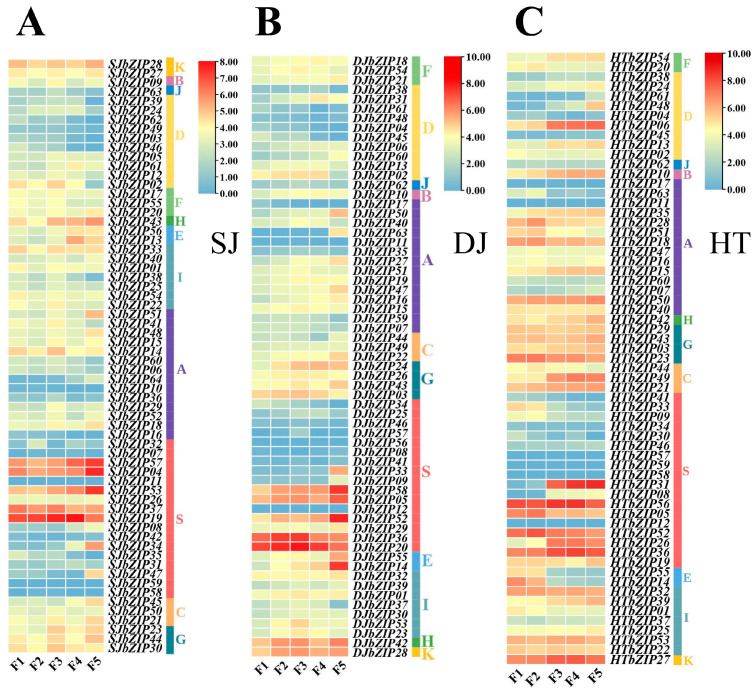
Expression trend of *bZIP* genes in flowering stages of jasmine. (**A**) Expression trend of *SJbZIP* genes. (**B**) Expression trend of *DJbZIP* genes. (**C**) Expression trend of *HTbZIP* genes.

**Figure 9 ijms-25-00488-f009:**
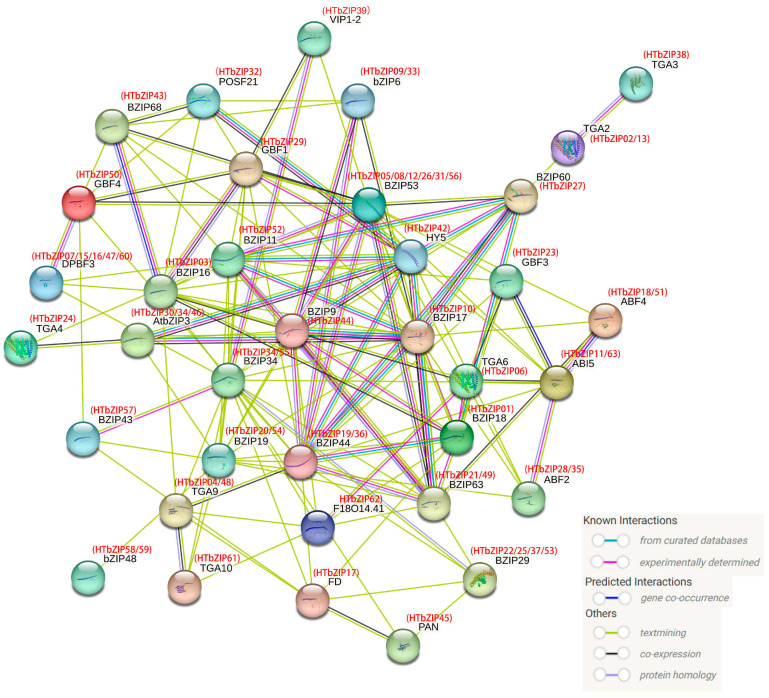
Protein–protein interaction network of HTbZIP proteins. Inside the nodes was a three-dimensional structure diagram of the protein, and the different colored lines between the nodes indicate the type of interaction evidence.

**Figure 10 ijms-25-00488-f010:**
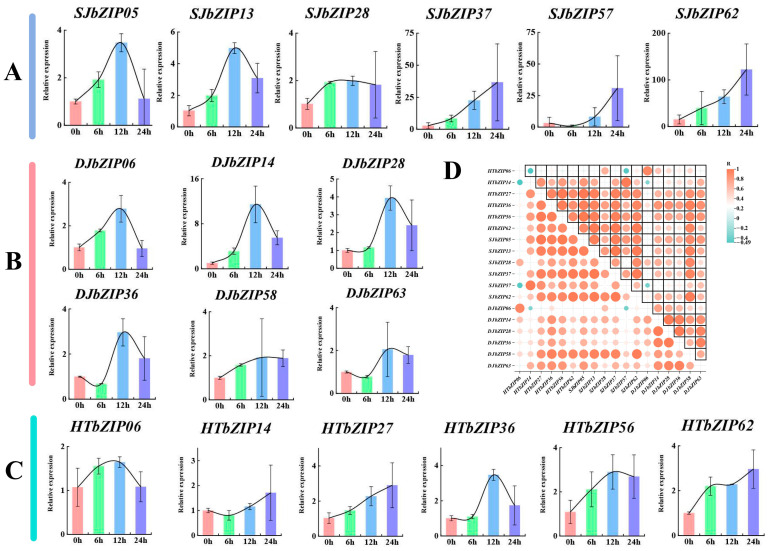
Jasimine *bZIP*s expression at 0 h, 6 h, 12 h and 24 h after 0.5 mg/L ABA treatment. (**A**) The expression of *SJbZIP* genes by ABA treatment in single-petal jasmine. (**B**) The expression of *DJbZIP* genes by ABA treatment in double-petal jasmine. (**C**) The expression of *HTbZIP* genes by ABA treatment in hutou jasmine. (**D**) The correlation heatmap of *bZIP* genes in three jasmines.

## Data Availability

The original HT jasmine genome described in this article have been deposited to NGDC. All data generated or analyzed during this study are included in this published article and are also available from the corresponding author on reasonable request.

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
