# Peer review of "The bZIP Transcription Factors in Current Jasmine Genomes: Identification, Characterization, Evolution and Expressions"

_ijms, 2023, doi:10.3390/ijms25010488_

Round 1

Reviewer 1 Report

Comments and Suggestions for Authors

see PDF

Author Response

Thank you for your review and valuable suggestions. Besides the following revision as your suggestions, we have modified the whole MS and the main expression logic is sorted out. We also supplemented some experiments.Hope this make our research more complete under your guidance, if there still any problems, please do not hesitate to contect with us.

Line 19:We have modified it to ABRE.

Line 43:We have reworked expression this of the paragraphs. The bZIP gene is widely distributed among species, and the image caption provided below is from PlantTFDB(http://planttfdb.gao-lab.org/family.php?fam=bZIP,accessed on 08 December 2023).

Line 44:For this paragraph, we have reworked the content and the “...a fixed...”is .....

Line 54: We have changed "α-helix" to "α-helix" .

Line 55: We have changed "homologous" to "homodimers" .

Line 59 and Line 67: We have revised your question.

Line 127, Line 214, Line 223, and Lines 232-235: Your comments are very insightful, and we have made revisions to each of the results.

Line 258: Thank you for your review. We have modified it.

Line 262 and Line 266: Thank you for your comments. We aim to compare the number of bZIP genes in jasmine with the number of bZIP genes in other species. The data presented represents the number of bZIP genes identified in other species.

Lines 297-99: ENC-plot analysis is used to observe the distance between the actual ENC and the expected value of each gene. If the gene locus fits the standard curve, it means that the codon use preference is affected by mutations. Otherwise, the codon use preference is mainly affected by natural factors [1]. In the Neutrality-plot analysis, if there is a significant correlation between GC12 and GC3, and the slope of the regression line is close to 1, it indicates that there is no significant difference in the base composition of the three positions. In this case, the codon bias is primarily influenced by mutations. If the correlation between GC12 and GC3 is not significant, it indicates that there are differences in the usage patterns of the 1st, 2nd, and 3rd bases, and the choice of codon is the main factor affecting codon preference[2]. [1]Yang M, Liu J, Yang W, Li Z, Hai Y, Duan B, Zhang H, Yang X, Xia C. Analysis of codon usage patterns in 48 Aconitum species. BMC Genomics. 2023 Nov 22;24(1):703. doi: 10.1186/s12864-023-09650-5. PMID: 37993787; PMCID: PMC10664653.[2]HE Yaling, PENG Yejun, LI Jin, FENG Bin, QING Yujie, WANG Aiying, ZHU Jianbo. Preference analysis of codon usage in the chloroplast genome of Saussurea involucrate. Journal of Shihezi University. 2022, 40(1): 84-92 https://doi.org/10.13880/j.cnki.65-1174/n.2022.22.009

Lines 300-311: Thank you for your comments, we have made changes in Lin396-403.

Line 317: We have modified it.

Line 319: Thank you for your review. It based on disclosing the novel function of the low-energy activated group S1 bZIP11-related TFs as regulators of auxin-mediated primary root growth. Whereas transgenic gain-of-function approaches of these bZIPs interfere with the activity of the root apical meristem and result in root growth repression, root growth of loss-of-function plants show a pronounced insensitivity to low-energy conditions[1]. [1].C, W., et al., The Arabidopsis bZIP11 transcription factor links low-energy signalling to auxin-mediated control of primary root growth. PLoS Genet, 2017. 13: p. e1006607.

Line 323: Thank you for your comments.The bud means flower bud and we have made changes and additions, Line 414-421.

Lines 328-30: Thanks for your comments, we've refined it. Line 432-435.

Reviewer 2 Report

Comments and Suggestions for Authors

The authors have presented the results and explained them well in this manuscript. However, I have a few comments to improve before acceptance.

1. Ln 103, Fig 1, legend is not self-explanatory, even though I did not see any explanation for the B. Please rewrite the legends

2. ln 141, why fig 3 is before fig 2, any reason? 2E legend is not there! make them self-explanatory.

3. Ln 158, 4B legend explanation is needed. remove one dot

4. Fig 6, B legend is before A, could you please explain it?

In the methods and materials, paragraphs 4.4, 4.5, 4.7 and 4.6 are too short and summery. I like to see a bit more explanation there.

Comments on the Quality of English Language

I am looking forward to seeing the revised file. Thank you.

Author Response

Thank you very much for your valuable comments on our articles. After careful consideration, we have modified and adjusted the whole MS, and added some experiments during the period, hoping that this version will be better

  1. Ln 103, Fig 1, legend is not self-explanatory, even though I did not see any explanation for the B. Please rewrite the legends

R1:Thank you for the comments. We accept your comments and make changes,Line101-105.

  1. ln 141, why fig 3 is before fig 2, any reason? 2E legend is not there! make them self-explanatory.

R2:Thanks for the correction. We have fixed the error issue.

  1. Ln 158, 4B legend explanation is needed. remove one dot

R3:We have supplemented the legend with that. Figure 4.

  1. Fig 6, B legend is before A, could you please explain it?

R4:Thank you for the comments. We accept your comments and make changes, in Fiure 6, Line 253-257.

  1. In the methods and materials, paragraphs 4.4, 4.5, 4.7 and 4.6 are too short and summery. I like to see a bit more explanation there.

R5:We have included additional information regarding the experimental methods in paragraphs 4.4, 4.5, 4.6 and 4.7.

Reviewer 3 Report

Comments and Suggestions for Authors
  1. Provide more specific and constructive feedback on the major issues identified. For example, suggest what additional analyses or experiments could address concerns about functional roles of bZIP genes. Give concrete recommendations.
  2. Offer guidance on strengthening the bioinformatics methodology where needed. For instance, are there any additional approaches or databases that could support the computational analyses performed?
  3. Use clear but polite language to critique issues with the writing or figures. Avoid overly harsh judgments. Maintain a collegial tone.
  4. Expand on what is done well in the paper and highlight unique contributions made to the field. Counterbalance criticism with positive feedback.
  5. Check that all cited gaps or issues link logically back to a clear recommendation for improvement. The roadmap for revisions should flow naturally.
  6. Supplement generalized statements about grammar, wording, or clarity with 1-2 illustrative examples of problematic phrasing. Be specific.
  7. Close by emphasizing that your suggestions are intended to move the manuscript toward publication by addressing problematic aspects while also building on existing strengths. Keep the recommendations constructive.
Comments on the Quality of English Language

no

Author Response

Thank you for your review and valuable suggestions. Besides the following revision as your suggestions, we have modified the whole MS and also supplemented some experiments.Hope this make our research more complete under your guidance, if there still any problems, please do not hesitate to contect with us

1.Provide more specific and constructive feedback on the major issues identified. For example, suggest what additional analyses or experiments could address concerns about functional roles of bZIP genes. Give concrete recommendations.

R1:Thank you for your review. We supplemented the Protein-protein interaction network, Predicted miRNA target distribution, and real-time fluorescence PCR analysis of the bZIP genes.

  1. Offer guidance on strengthening the bioinformatics methodology where needed. For instance, are there any additional approaches or databases that could support the computational analyses performed?

R2:Thank you for your suggestion, we have supplement more details on the methodology.

  1. Use clear but polite language to critique issues with the writing or figures. Avoid overly harsh judgments. Maintain a collegial tone.

R3: Thank you for your comment. We have made changes to the writing of the article.

  1. Expand on what is done well in the paper and highlight unique contributions made to the field. Counterbalance criticism with positive feedback.

R4:Thank you for your suggestions, we have modified the whole MS, especially highlight unique results in the research we found.

  1. Check that all cited gaps or issues link logically back to a clear recommendation for improvement. The roadmap for revisions should flow naturally.

R5:Thank you, we have checked.

  1. Supplement generalized statements about grammar, wording, or clarity with 1-2 illustrative examples of problematic phrasing. Be specific.

R6:Thank you for your patience and careful checking for the MS.

  1. Close by emphasizing that your suggestions are intended to move the manuscript toward publication by addressing problematic aspects while also building on existing strengths. Keep the recommendations constructive.

R7:Thank you for your suggestions, we have modified the whole MS according to your suggestion, hope this version is better.

Round 2

Reviewer 1 Report

Comments and Suggestions for Authors

I'm in agree with modifications. Congratulations to the authors.

Reviewer 2 Report

Comments and Suggestions for Authors

Looking better. Thanks 

Reviewer 3 Report

Comments and Suggestions for Authors

I don't have no more comments